# Antimicrobial Activity of Propolis from the Brazilian Stingless Bees *Melipona quadrifasciata anthidioides* and *Scaptotrigona depilis* (Hymenoptera, Apidae, Meliponini)

**DOI:** 10.3390/microorganisms11010068

**Published:** 2022-12-26

**Authors:** Jaqueline Ferreira Campos, Thaliny Bonamigo, Paola dos Santos da Rocha, Vanessa Marina Branco Paula, Uilson Pereira dos Santos, José Benedito Perrella Balestieri, Denise Brentan Silva, Carlos Alexandre Carollo, Leticia M. Estevinho, Kely de Picoli Souza, Edson Lucas dos Santos

**Affiliations:** 1Research Group on Biotechnology and Bioprospecting Applied to Metabolism (GEBBAM), Federal University of Grande Dourados, Dourados 79804-970, Brazil; 2CIMO-Mountain Research Center, Department of Biology and Biotechnology, Polytechnic Institute of Bragança, Campus Santa Apolónia, Agricultural College of Bragança, Bragança 5301-855, Portugal; 3Laboratory of Natural Products and Mass Spectrometry, Federal University of Mato Grosso do Sul, Cidade Universitária, Campo Grande 79070-900, Brazil

**Keywords:** natural products, Meliponini, HPLC-ESI-MS, resistant microorganisms

## Abstract

*Melipona quadrifasciata anthidioides* and *Scaptotrigona depilis* are species of stingless bees capable of producing propolis, which has considerable bioprospecting potential. In this context, the objective of this study was to determine the chemical compositions and evaluate the antimicrobial activity of propolis produced by *M. q. anthidioides* and *S. depilis*. The ethanolic extracts of propolis of *M. q. anthidioides* (EEP-M) and *S. depilis* (EEP-S) were prepared, and their chemical constituents were characterized by HPLC-ESI-MS. The antimicrobial activity was evaluated against bacteria and fungi, isolated from reference strains and hospital origin resistant to the action of antibiotics. From EEP-M, phenolic compounds were annotated, including gallic acid, ellagic acid, and flavonoids, as well as diterpenes and triterpenes. EEP-S showed mainly triterpene in its chemical composition. Both extracts inhibited the growth of medically relevant bacteria and fungi, including hospital-acquired and antimicrobial-resistant. In general, EEP-S showed better antimicrobial activity compared to EEP-M. The MIC of EEP-S against vancomycin-resistant *Enterococcus faecalis* was 3.50 mg/mL, while the MIC of EEP-M was 5.33 ± 0.16 mg/mL. In conclusion, this study shows that propolis produced by *M. q. anthidioides* and *S. depilis* has the potential to be used for the prevention or treatment of microbial infections.

## 1. Introduction

*Melipona quadrifasciata anthidioides* (Lepeletier, 1836) and *Scaptotrigona depilis* (Moure, 1942) are species of stingless bees found in South America, distributed in Argentina, Paraguay, Bolivia, and Brazil [1]. These bees belong to the Meliponini tribe and are efficient pollinators of native plants [2]. Additionally, they can produce honey as a nutritional source for offspring in addition to cerumen and propolis, which provide mechanical and biological protection to the bees of the hive [3].

Among bee products, propolis has been widely studied because it is a complex bioactive mixture known for its high chemical diversity [3,4,5] and important pharmacological activities [6,7]. Propolis is formed by mixing plant exudates with salivary enzymes from bees, resulting in a viscous material with variable color and flavor [8,9].

These unique characteristics render propolis a product of commercial interest and great pharmacological potential, since qualitative and quantitative changes in the chemical compounds found in propolis modify its therapeutic properties [10,11,12]. Some studies describe the chemical composition of propolis from *M. q. anthidioides* and *S. depilis*, reporting a predominance of diterpenes [13,14] in addition to phytosterols, phenolic compounds, and tocopherol [15]. These compounds may be related to the biological activities already described for these products, such as antibacterial [9,13], antioxidant [14,15], and cytotoxic activities [15].

Given the therapeutic potential of the propolis from *M. q. anthidioides* and *S. depilis*, this study aimed to investigate the chemical composition of propolis from these species and evaluate its antimicrobial activity against different bacteria and yeasts, isolated from reference strains and hospital origin resistant to the action of antibiotics.

## 2. Materials and Methods

### 2.1. Preparation of the Ethanol Extract of Propolis

Propolis samples from *M. q. anthidioides* and *S. depilis* were collected from the state of Mato Grosso do Sul (22°13′12″ S–54°49′2″ W), in the Midwest region of Brazil, with a total of seven collections being performed for each species. The ethanol extract of propolis was prepared according to the method described by Bonamigo et al. [15], using 4.5 mL of 80% ethanol per 1 g of propolis. The extraction was performed at 70 °C until total dissolution, and, subsequently, this material was filtered by filter paper qualitative 80 g/m^2^ (Prolab, São Paulo, SP, Brazil) to obtain the ethanolic extracts of propolis of *M. q. anthidioides* (EEP-M) and *S. depilis* (EEP-S). After the preparation of the extracts, they were kept at a temperature of −20 °C until analysis.

### 2.2. Analyses by High-Performance Liquid Chromatography Coupled to Diode Array Detector and Mass Spectrometry (HPLC-DAD-MS)

Five microliters of each sample, EEP-M or EEP-S (1 mg/mL), were injected into an LC-20AD ultra-fast liquid chromatograph (UFLC) (Shimadzu) coupled to a diode array detector (DAD) and a mass spectrometer micrOTOF-Q III (Bruker Daltonics) with electrospray ionization source (ESI) and quadrupole and time-of-flight analyzers. A column Kinetex C-18 (150 mm × 2.2 mm inner diameter, 2.6 μm) was used in the analyses and maintained at 50 °C during the analyses. The mobile phase consisted of deionized water (A) and acetonitrile (B), both containing 0.1% formic acid, and the following elution gradient profile was applied: 0–2 min-3% B; 2–25 min-3–25% B; 25–40 min-25–80% B; and 40–43 min-80% B. The gradient was followed by reconditioning of the column (5 min). The flow rate was 0.3 mL/min. The samples were analyzed in negative and positive ion mode (*m*/*z* 120–1300). Nitrogen was applied as a nebulizer (4 Bar), drying (9 mL/min), and collision gas. The capillary voltage was 4500 kV.

### 2.3. Antimicrobial Activity

The antimicrobial activity of EEP-M and EEP-S was investigated in microorganisms collected from biological fluids at the Hospital Center and identified in the Microbiology Laboratory of Escola Superior Agrária (ESA) de Bragança, Portugal. Reference strains were obtained from the authorized ATCC distributor (LGC Standards SLU, Barcelona, Spain), as listed in Table 1.

The microorganisms were stored in a Mueller–Hinton broth supplemented with 20% glycerol at −70 °C before experimental use. The inoculum was then prepared by dilution of the cell mass in 0.85% NaCl solution, adjusted to 0.5 on the MacFarland scale, as confirmed by spectrophotometric readings at 580 and 640 nm, for bacteria and yeast, respectively. Antimicrobial assays were performed as described by Silva et al. [16] using nutrient broth (NB) for bacteria or yeasts peptone dextrose (YPD) for yeast in microplates of 96 wells. The extracts were diluted in dimethylsulfoxide (DMSO) and transferred to the first well, followed by serial dilution (0.625–160 mg/mL). The inoculum was added to all wells (10^4^ colony forming units (CFU)/mL), and the plates were incubated at 37 °C for 24 h for bacteria and 25 °C for 48 h for yeast. Media controls were conducted with and without inoculum, and 0.27% DMSO alone was used as a solvent control in the inoculated medium. In addition, gentamicin and amphotericin B were used as antibacterial and antifungal positive controls, respectively. After the incubation period, the antimicrobial activity was detected by the addition of 20 μL of 2,3,5-triphenyl-2H-tetrazolium chloride (TTC) solution (5 mg/mL). The minimum inhibitory concentration (MIC) was defined as the lowest concentration of EEP-M and EEP-S that visibly inhibited the growth of microorganisms, as indicated by TTC staining, which marks viable cells in red color, due to the formation of formazan. To determine the minimum bactericidal concentration (MBC) and minimum fungicidal concentration (MFC), 20 μL of the last well where growth was observed and from each well where no color changes were seen was seeded in NB or YPD and incubated for 24 h at 37 °C for bacteria growth and 48 h for yeast growth. The lowest concentration that did not result in growth (<10 CFU/plate) after this subculture process was considered the MBC or MFC. The experiments were performed in triplicate, and the results were expressed in mg/mL. The data are shown as the mean ± standard error of the mean (SEM).

### 2.4. Statistical Analysis

Statistical analysis was performed for statistically significant differences between groups using one-way analysis of variance (ANOVA) followed by the Newman–Keuls test for the comparison of more than two groups using the Prism 5 GraphPad Software (GraphPad Software Inc., San Diego, CA, USA). The results were considered significant when *p* < 0.05.

## 3. Results

### 3.1. Chemical Composition by HPLC-DAD-MS

The extracts EEP-M and EEP-S were analyzed by HPLC-DAD-MS, and their constituents could be identified by UV, MS (accurate mass), and MS/MS data compared with data reported in the literature. The molecular formulas were determined considering errors and m-Sigma up 8 ppm and 30, respectively. In addition, some compounds were confirmed by injection of authentic standards. Thus, forty-seven compounds were detected and summarized in Table 2, and the chromatograms are illustrated in Figure 1. Chemical differences between EEP-M and EEP-S were evidenced, such as the presence of nonpolar compounds in EEP-S, which are not present in EEP-M. Additionally, EEP-M revealed mainly phenolic compounds in its composition.

Compounds **5** and **6** were confirmed by injection of authentic standards and identified as gallic acid and ellagic acid, respectively. In addition, peaks **1**–**4** revealed an absorption band with λ_max_ at 270 nm in their UV spectra, which is compatible with the chromophore of gallic acid [17]. For these components, the fragment ions at *m*/*z* 169 were observed, indicating the presence of galloyl substituent, while the ion *m*/*z* 301 suggested the hexahydroxydiphenoyl group. These components were annotated as hydrolysable tannins *O*-galloyl hexoside (**1**), di-*O*-galloyl hexoside (**2** and **4**), and *O*-galloyl- hexahydroxydiphenoyl hexoside (**3**). Their spectral data are compatible with the data described in the literature [17,18].

The compounds **7**–**8**, **10**, **15**–**16**, **18**–**19**, and **27** showed two absorption bands at the wavelength ≈280 and 310 nm, which are compatible and suggested, together with MS/MS data, the chromophores relative to galloyl and coumaroyl substituents [19]. Beyond fragment ions at *m*/*z* 169 [gallic acid-H]-, losses of 146 or 164 *u* (146 + H_2_O) suggested the coumaroyl substituents [17]. These metabolites were putatively annotated as *O*-coumaroyl *O*-galloyl hexoside (**7** and **8**), *O*-coumaroyl di-*O*-galloyl hexoside (**10**), *O*-coumaroyl tetra-*O*-galloyl hexoside (**15**), di-*O*-coumaroyl hexoside (**16**), di-*O*-coumaroyl *O*-galloyl hexoside (**18**), *O*-coumaroyl *O*-galloyl *O*-benzoyl hexoside (**19**), and *O*-coumaroyl *O*-cynnamoyl *O*-galloyl hexoside (**27**). The compounds **13** and **14** also showed losses of 148 *u* relative to losses of a cinnamoyl and subsequently a water molecule, as reported by Jin et al. [20], and they were annotated as *O*-cinnamoyl *O*-galloyl hexoside (**13**) and *O*-cinnamoyl di-*O*-galloyl hexoside (**14**).

The chromatographic peaks **9**, **17**, **21**, and **28** presented UV spectra (λ_max_ ≈ 290 and 330 nm—shoulder) compatible with flavanones [19]. The MS/MS data were compared to fragmentations reported in the literature, and they revealed relevant fragments to annotate them such as losses of CO, retro-Diels–Alder fission of the C ring, and radical methyl [21,22]. Thus, these components were annotated as eriodictyol (**9**), naringenin (**17**), *O*-methyl eriodictyol (**21**), and *O*-methyl naringenin (**28**) [19,22,23].

The compounds **31**–**34** and **39** revealed deprotonated ions compatible with a molecular formula that suggested diterpenes, while **37**–**38** and **41** were similar for triterpenes. The compound **39** revealed a fragmentation pathway similar to the diterpene abietic acid, which is a component already described from propolis of *M. quadrifasciata* [21].

### 3.2. Antimicrobial Activity

Investigation of the antimicrobial activity of the propolis extracts of *M. q. anthidioides* and *S. depilis* revealed both to be effective against the microorganisms evaluated; EEP-S was more effective than EEP-M. Inhibitory and bactericidal activity against gram-positive and gram-negative bacteria were observed, including hospital-acquired strains resistant to methicillin and vancomycin (Table 3). The extracts also showed inhibitory and fungicidal activity against *Cryptococcus neoformans* and *Candida albicans*, in both reference strains and amphotericin-B-resistant strains (Table 4).

## 4. Discussion

Propolis is a bee product known for centuries for its medicinal properties, including its antiseptic, healing, anti-inflammatory, and anticancer properties [3,5,24]. These activities are related to the chemical composition of propolis, which varies according to the local vegetation, season, and bee species that generate this product [25,26,27]. In this study, the chemical composition of propolis from stingless bees *M. q. anthidioides* and *S. depilis* varied among the evaluated samples. The extracts showed bactericidal and fungicidal activity against reference strains and hospital origin resistant to the action of antimicrobial agents.

The EEP-M presented in its composition 46 compounds, among them phenolic compounds, including gallic acid, ellagic acid, and flavonoids such as naringenin and eriodictyol. In addition, the EEP-M presented triterpenes, which were also detected in the EEP-S. Interestingly, EEP-S showed still unknown lipophilic compounds, which ratifies bee products as sources of new bioactive molecules, since this extract has proved to be a more potent antimicrobial in inhibiting the growth of medically relevant microorganisms, including the bacteria *Staphylococcus aureus* and *Pseudomonas aeruginosa* and the yeast *C. albicans*.

Przybyłek and Karpinski [9] reported that propolis promotes antibacterial activity by increasing the permeability of the cell membrane, disruption of membrane potential and adenosine triphosphate (ATP) production, and by decreasing bacterial motility. These mechanisms of action of propolis are correlated with the chemical profile, which may correspond to the different proportions of terpenes and phenolic compounds.

Lipophilic compounds such as terpenes, present in EEP-M and EEP-S, are described in the literature because they present antimicrobial action [28,29].

Cornara et al. [25] emphasized that the antimicrobial activity of different samples of propolis is related to the presence of terpenes such as α-pinene, β-pinene, δ-cadinene, farnesol, and dihydroeudesmol. Terpenes can cross the cell membrane and promote the loss of essential intracellular components, resulting in the death of microorganisms such as bacteria and fungi [30].

In addition to terpenes, in other studies with propolis extracts, antimicrobial activity against different strains of Staphylococcus was attributed to the presence of phenolic compounds such as caffeic acid and its derivatives and flavonoids such as pinostrobin, pinocembrin, chrysin, and galangin [31].

Phenolic compounds as flavonoids can act by inhibiting the activity of the enzymes RNA polymerase [25], DNA gyrase, and ATP synthase and by inhibiting virulence factors such as lipopolysaccharides present in the outer membrane of gram-negative bacteria [32]. Flavonoids are the largest group of phenolic compounds, totaling approximately 6500 compounds [33], and are widely known for their biological activities.

Additionally, flavonoids identified in different propolis extracts, such as quercetin, myricetin, kaempferol, pinocembrin, and naringenin, have antifungal activity against *Candida* spp., acting mainly in the inhibition of the development of this microorganism [34]. Haghdoost et al. [35] reported that propolis decreases the formation of germ tubes, one of the main virulence factors of fungi, such as *C. albicans*.

Gucwa et al. [36] reported the antifungal action of Polish propolis extract and attributed the depolarization of the fungal membrane and inhibition of hyphae formation in *C. albicans* as the main mechanisms of action. The authors also highlight that of the 50 propolis samples evaluated, the ones with the highest antifungal activity had higher flavones and flavonols content than extracts with the lowest antifungal activity [36].

In conclusion, this study demonstrates that despite their very different compositions, propolis extracts produced by both *M. q. anthidioides* and *S. depilis* stingless bees were active, showing that these bee products have the potential to be used for the prevention or treatment of microbial infections.

## Figures and Tables

**Figure 1 microorganisms-11-00068-f001:**
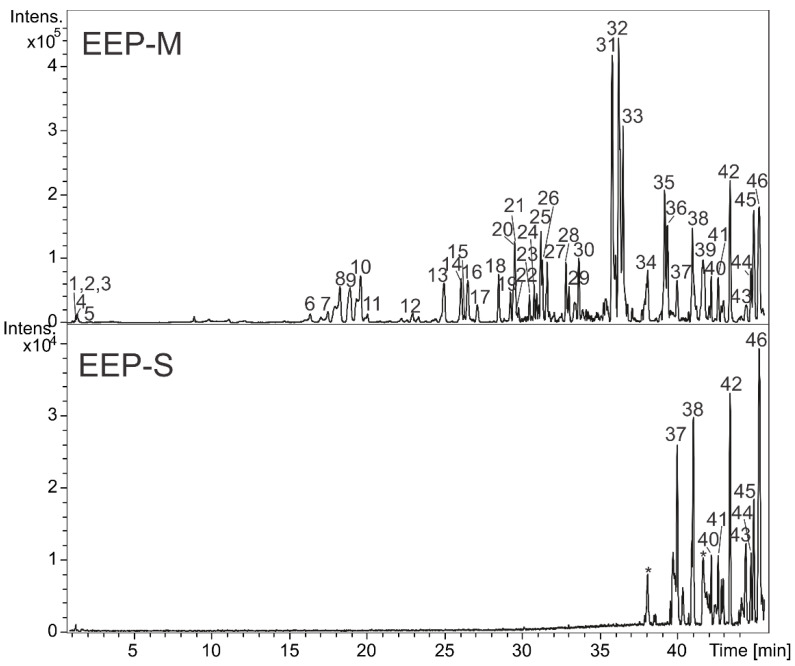
Base peak chromatogram (negative ion mode) from ethanolic extracts of *Melipona quadrifasciata anthidiodes* (EEP-M) and *Scaptotrigona depilis* (EEP-S) propolis by LC-DAD-MS. (* contaminant peaks from the chromatographic system.)

**Table 1 microorganisms-11-00068-t001:** Strains of microorganisms used to test the antimicrobial activity of EEP-M and EEP-S.

Microorganisms	Reference	Origin
Bacteria		
*Staphylococcus aureus*	ATCC^®^ 6538™	Reference culture
Methicillin-resistant *Staphylococcus aureus*	ESA 175	Pus
Methicillin-resistant *Staphylococcus aureus*	ESA 159	Expectoration
*Enterococcus faecalis*	ATCC^®^ 43300™	Reference culture
Vancomycin-resistant *Enterococcus faecalis*	ESA 201	Urine
Vancomycin-resistant *Enterococcus faecalis*	ESA 361	Rectal swabs
*Escherichia coli*	ATCC^®^ 29998™	Reference culture
Cephalosporin-resistant *Escherichia coli*	ESA 37	Urine
Cephalosporin-resistant *Escherichia coli*	ESA 54	Hemoculture
*Pseudomonas aeruginosa*	ATCC^®^ 15442™	Reference culture
Imipenem-resistant *Pseudomonas aeruginosa*	ESA 22	Expectoration
Imipenem-resistant *Pseudomonas aeruginosa*	ESA 23	Gingival exudates
**Fungi**		
*Cryptococcus neoformans*	ATCC^®^ 32264	Reference culture
Amphotericin B-resistant *Cryptococcus neoformans*	ESA 211	Blood
Amphotericin B-resistant *Cryptococcus neoformans*	ESA 105	Skin biopsy
*Candida albicans*	ATCC^®^ 10231™	Reference culture
Amphotericin B-resistant *Candida albicans*	ESA 100	Feces
Amphotericin B-resistant Candida albicans	ESA 97	Urine

**Table 2 microorganisms-11-00068-t002:** Chemicals constituents identified from ethanolic extracts of *Melipona quadrifasciata anthidiodes* (EEP-M) and *Scaptotrigona depilis* (EEP-S) propolis by LC-DAD-MS.

Peak	RT (min)	UV (nm)	Molecular Formula	[M-H]^-^ (*m*/*z*)	MS/MS (*m*/*z*)	Compound	EEP-M	EEP-S
1	1.2	270	C_13_H_16_O_10_	331.0677	169	*O*-galloyl hexoside	+	-
2	1.2	270	C_20_H_20_O_14_	483.0781	169	di-*O*-galloyl hexoside	+	-
3	1.2	270	C_27_H_22_O_18_	633.0749	301, 275, 249, 169	*O*-galloyl-HHDP hexoside	+	-
4	1.3	270	C_20_H_20_O_14_	483.0782	169	di-*O*-galloyl hexoside	+	-
5	2.2	269	C_7_H_6_O_5_	169.0127	-	Gallic acid ^st^	+	-
6	16.4	254, 366	C_14_H_6_O_8_	300.9990	245, 229	Ellagic acid ^st^	+	-
7	17.5	283, 310	C_22_H_22_O_12_	477.1038	331, 313, 271, 241, 169	*O*-coumaroyl *O*-galloyl hexoside	+	-
8	18.3	290, 310	C_22_H_22_O_12_	477.1054	331, 313, 265, 205, 169	*O*-coumaroyl *O*-galloyl hexoside	+	-
9	18.9	289, 333 (sh)	C_15_H_12_O_6_	287.0571	259, 277, 173	Eriodictyol	+	-
10	19.3	286, 310	C_29_H_26_O_16_	629.1166	465, 459, 316, 295, 271, 211, 169	*O*-coumaroyl di-*O*-galloyl hexoside	+	-
11	20.1	278	C_20_H_20_O_11_	435.0950	169	Gallic acid derivative	+	-
12	22.9	281, 308 (sh)	C_20_H_24_O_6_	359.1502	329, 159	Unknown	+	-
13	24.9	282	C_22_H_22_O_11_	461.1088	313, 253, 211, 189, 169, 161	*O*-cinnamoyl *O*-galloyl hexoside	+	-
14	26.1	279	C_29_H_26_O_15_	613.1214	465, 313, 271, 211, 169	*O*-cinnamoyl di-*O*-galloyl hexoside	+	-
15	26.3	281, 308	C_43_H_34_O_24_	933.1368	615, 169	*O*-coumaroyl tetra-*O*-galloyl hexoside	+	-
16	26.5	300, 312	C_24_H_24_O_10_	471.1292	307, 265, 205, 187, 163, 145	di-*O*-coumaroyl hexoside	+	-
17	27.1	288, 325 (sh)	C_15_H_12_O_5_	271.0607	151	Naringenin	+	-
18	28.5	290, 311	C_31_H_28_O_14_	623.1412	477, 459, 313, 271, 169	di-*O*-coumaroyl *O*-galloyl hexoside	+	-
19	29.3	292, 310	C_29_H_26_O_13_	581.1310	417, 187, 169, 163	*O*-coumaroyl *O*-galloyl *O*-benzoyl hexoside	+	-
20	29.4	288, 310	C_22_H_22_O_9_	429.1196	187, 163, 145	Coumaric acid derivative	+	-
21	29.5	290, 335 (sh)	C_16_H_14_O_6_	301.0726	273, 258, 179, 165	*O*-methyl eriodictyol	+	-
22	29.8	286	C_23_H_20_O_7_	407.1141	313, 285, 245, 201, 177	Unknown	+	-
23	30.5	288, 320 (sh)	C_31_H_30_O_13_	609.1642	581, 441, 307, 283, 273, 179	Unknown	+	-
24	30.8	280, 320 (sh)	C_22_H_26_O_7_	401.1615	326, 205, 190	Unknown	+	-
25	31.1	284, 315	C_24_H_24_O_9_	455.1369	187, 163, 145	*O*-coumaroyl *O*-cynamoyl hexoside	+	-
26	31.4	292	C_23_H_20_O_7_	407.1161	313, 285, 245, 203, 177, 151	Unknown	+	-
27	31.7	281, 312	C_31_H_28_O_13_	607.1485	461, 443, 313, 271, 211, 169	*O*-coumaroyl *O*-cinnamoyl *O*-galloyl hexoside	+	-
28	32.9	286, 328 (sh)	C_16_H_14_O_5_	285.0788	165	*O*-methyl naringenin	+	-
29	33.1	289	C_24_H_22_O_7_	421.1320	393, 363, 299, 271, 165	Unkown	+	-
30	33.7	295	C_24_H_22_O_7_	421.1328	393, 363, 299, 285, 271, 179, 165	Unkown	+	-
31	35.9	272	C_20_H_32_O_3_	319,2313	-	Diterpene	+	-
32	36.2	275	C_20_H_32_O_3_	319.2314	-	Diterpene	+	-
33	36.2	275	C_20_H_32_O_3_	319.2314	-	Diterpene	+	-
34	38.1	284	C_20_H_28_O_2_	299.2037	-	Diterpene	+	-
35	39.2	-	C_22_H_34_O_4_	365.2405	301	Unknown	+	-
36	39.4	284	C_21_H_28_O_3_	327.1987	312, 297, 201	Unknown	+	-
37	40.0	-	C_30_H_48_O_4_	471.3494	453, 441, 427, 407	Triterpene	+	+
38	41.1	-	C_30_H_46_O_4_	469.3337	451, 439, 421, 407	Triterpene	+	+
39	41.7	254	C_20_H_30_O_2_	301.2184	283, 229	Abietic acid	+	-
40	42.2	275	C_23_H_34_O_2_	341.2499	299, 191	Unknown	+	+
41	42.7	-	C_30_H_48_O_4_	471.3467	425, 357	Triterpene	+	+
42	43.4	276	C_23_H_36_O_2_	343.2653	301, 285	Unknown	+	+
43	44.5	-	C_31_H_50_O_3_	469.3676	-	Unknown	+	+
44	44.8	275	C_21_H_36_O_2_	319.2649	277	Unknown	+	+
46	44.9	-	C_24_H_34_O_3_	369.2421	325	Unknown	+	+
47	48.3	275	C_23_H_38_O_2_	345.2801	303	Unknown	+	+

RT: retention time; HHDP: hexahydroxydiphenoyl; st: confirmed by authentic standard; sh: shoulder; +: present; -: absent.

**Table 3 microorganisms-11-00068-t003:** Minimum inhibitory concentration (MIC) and minimum bactericidal concentration (MBC) for the studied bacteria, gram-negative and gram-positive.

Microorganisms (Bacteria)	EEP-M (mg/mL)		EEP-S (mg/mL)		Gentamicin (μg/mL)	
	MIC	MBC	MIC	MBC	MIC	MBC
*Staphylococcus aureus* ATCC^®^ 6538™	3.00 ± 0.14 ^a^	4.33 ± 0.22 ^A^	1.67 ± 0.17 ^b^	2.25 ± 0.14 ^B^	1.67 ± 0.17 ^c^	2.00 ± 0.29 ^C^
Methicillin-resistant *Staphylococcus aureus* ESA 175	3.58 ± 0.30 ^a^	5.00 ± 0.14 ^A^	2.00 ± 0.29 ^b^	3.08 ± 0.08 ^B^	1.83 ± 0.17 ^c^	2.67 ± 0.17 ^C^
Methicillin-resistant *Staphylococcus aureus* ESA 159	3.92 ± 0.08 ^a^	5.50 ± 0.28 ^A^	2.67 ± 0.17 ^b^	4.17 ± 0.17 ^B^	2.00 ± 0.29 ^c^	2.50 ± 0.29 ^C^
*Enterococcus faecalis* ATCC^®^ 43300™	4.75 ± 0.54 ^a^	6.92 ± 0.22 ^A^	3.00 ± 0.29 ^b^	3.75 ± 0.14 ^B^	2.17 ± 0.17 ^c^	2.83 ± 0.30 ^C^
Vancomycin-resistant *Enterococcus faecalis* ESA 201	5.33 ± 0.16 ^a^	7.17 ± 0.44 ^A^	3.50 ± 0.29 ^b^	5.17 ± 0.17 ^B^	2.33 ± 0.17 ^c^	3.25 ± 0.14 ^C^
Vancomycin-resistant *Enterococcus faecalis* ESA 361	5.83 ± 0.44 ^a^	7.50 ± 0.52 ^A^	4.67 ± 0.17 ^a^	6.5 ± 0.29 ^A^	2.67 ± 0.17 ^b^	3.33 ± 0.17 ^B^
*Escherichia coli* ATCC^®^ 29998™	6.00 ± 0.30 ^a^	9.83 ± 0.44 ^A^	3.50 ± 0.29 ^b^	6.33 ± 0.17 ^B^	4.09 ± 0.08 ^c^	4.58 ± 0.30 ^C^
Cephalosporin-resistant *Escherichia coli* ESA 37	7.25 ± 0.14 ^a^	10.50 ± 0.29 ^A^	5.75 ± 0.14 ^b^	8.33 ± 0.33 ^B^	4.67 ± 0.17 ^c^	4.67 ± 0.22 ^C^
Cephalosporins-resistant *Escherichia coli* ESA 54	7.75 ± 0.14 ^a^	11.17 ± 0.22 ^A^	6.50 ± 0.29 ^b^	8.83 ± 0.44 ^B^	4.42 ± 0.08 ^c^	4.92 ± 0.08 ^C^
*Pseudomonas aeruginosa* ATCC^®^ 15442™	8.42 ± 0.30 ^a^	12.00 ± 0.50 ^A^	6.83 ± 0.17 ^b^	9.50 ± 0.38 ^B^	4.75 ± 0.14 ^c^	5.00 ± 0.29 ^C^
Imipenem-resistant *Pseudomonas aeruginosa* ESA 22	9.33 ± 0.33 ^a^	12.58 ± 0.30 ^A^	8.25 ± 0.38 ^a^	11.08 ± 0.08 ^B^	5.67 ± 0.17 ^b^	6.17 ± 0.17 ^C^
Imipenem-resistant *Pseudomonas aeruginosa* ESA 23	9.92 ± 0.68 ^a^	13.08 ± 0.30 ^A^	8.75 ± 0.43 ^a^	12.00 ± 0.29 ^A^	6.67 ± 0.33 ^b^	6.50 ± 0.29 ^B^

Values are expressed as mean ± SEM. N = 3 experiment per group. Different letters represent statistical differences between groups (*p* < 0.05): lowercase letters for MIC and uppercase letters for MBC.

**Table 4 microorganisms-11-00068-t004:** Minimum inhibitory concentration (MIC) and minimum fungicidal concentration (MFC) for the studied fungi.

Microorganisms (Fungi)	EEP-M (mg/mL)		EEP-S (mg/mL)		Amphotericin B (μg/mL)	
	MIC	MFC	MIC	MFC	MIC	MFC
*Cryptococcus neoformans* ATCC^®^ 32264	11.42 ± 0.30 ^a^	14.33 ± 0.44 ^A^	7.00 ± 0.29 ^b^	10.50 ± 0.29 ^B^	0.55 ± 0.03 ^c^	0.87 ± 0.07 ^C^
Amphotericin B-resistant *Cryptococcus neoformans* ESA 211	12.58 ± 0.30 ^a^	15.25 ± 0.14 ^A^	7.83 ± 0.17 ^b^	12.16 ± 0.17 ^B^	0.62 ± 0.06 ^c^	1.25 ± 0.14 ^C^
Amphotericin B-resistant *Cryptococcus neoformans* ESA 105	13.25 ± 0.14 ^a^	16.67 ± 0.54 ^A^	8.50 ± 0.57 ^b^	12.33 ± 0.17 ^B^	0.63 ± 0.02 ^c^	1.67 ± 0.22 ^C^
*Candida albicans* ATCC^®^ 10231™	14.25 ± 0.14 ^a^	18.42 ± 0.30 ^A^	8.50 ± 0.29 ^b^	13.00 ± 0.76 ^B^	0.72 ± 0.04 ^c^	0.92 ± 0.16 ^C^
Amphotericin B-resistant *Candida albicans* ESA 100	15.75 ± 0.38 ^a^	19.58 ± 0.30 ^A^	10.50 ± 0.29 ^b^	14.83 ± 0.17 ^B^	0.82 ± 0.04 ^c^	1.67 ± 0.08 ^C^
Amphotericin B-resistant Candida albicans ESA 97	16.50 ± 0.28 ^a^	20.75 ± 0.14 ^A^	11.67 ± 0.17 ^b^	16.00 ± 0.29 ^B^	0.92 ± 0.02 ^c^	1.75 ± 0.14 ^C^

Values are expressed as mean ± SEM. N = 3 experiment per group. Different letters represent statistical differences between groups (*p* < 0.05): lowercase letters for MIC and uppercase letters for MFC.

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
