# Peer review of "Antimicrobial Activity of Propolis from the Brazilian Stingless Bees *Melipona quadrifasciata anthidioides* and *Scaptotrigona depilis* (Hymenoptera, Apidae, Meliponini)"

_microorganisms, 2022, doi:10.3390/microorganisms11010068_

Round 1

Reviewer 1 Report

The study of Authors: Jaqueline Ferreira Campos, Thaliny Bonamigo, Paola dos Santos da Rocha, José Benedito Perrella Balestieri, Denise Brentan Silva, Carlos Alexandre Carollo, Leticia M. Estevinho, Kely de Picoli Souza, Edson Lucas dos Santos, entitled: “Antimicrobial activity of propolis from the Brazilian stingless bees Melipona quadrifasciata anthidioides and Scaptotrigona depilis (Hymenoptera, Apidae, Meliponini)”, is well suited for Communication section of Microorganisms MDPI journal. Authors aimed to compare the antibacterial and antifungal activity of the ethanol based- propolis extracts of two Brazilian stingless bees, employing variety of commercial microbes with and without resistance to the action of antibiotics. They also have employed HPLC-DAD-MS/MS to study chemical composition of propolis extracts. It was such a relief and I am so grateful for receiving a neat manuscript for review, with a proper English language style and acceptable grammar, so it was very easy to read and add a few of comments.

In spite of large body of literature data regarding the antimicrobial activity of ethanol/based propolis bees extracts, with their parallel chemical composition determination, as early as e.g. publication of Srdjan Stepanovic et  al 2003., “In vitro antimicrobial activity of propolis and synergism between propolis and antimicrobial drugs”, https://doi.org/10.1078/0944-5013-00215), I find this communication manuscript enough interesting for publication, due to interesting results of comparison of two bees propolis extracts, where EEP-S is very obscure in terms of differential compound scope in contrast to EEP-M, which is rich in numerous compounds however less active, in exerting antibacterial and antifungal effects.

Before being suitable for publication the following comments should be addressed

1.      As already mentioned, the results are the strongest part of this manuscript, e.g. its valuable novelty. Therefore they should be clearly communicated in at least conclusion part, if not in the respective results sections (please take care to follow word number limitation imposed by journal). For example, it is obvious that EEP-S has half a dozen compounds mostly of unknown and triterpene kind, while EEP-M has several dozens of compounds, mostly small phenolics and those contained in EEP-S but significantly less. Yet, EEP-S is more potent in killing and inhibiting the growth of bacteria and fungi. It is so interesting, don’t let others elaborate your results.

2.      What about statistical tests? How do we know that differences are significant?

3.      Please take care about numeration of Tables throughout the text. Table 3 appears as the first in the text, lines 56 and 57, then Table 4 at 88 line, etc…arrange these accordingly.

4.      How do you know that peaks shown in Figure 1 in EEP-S is contaminant peak, what is this compound?

5.      Perhaps you could insert little bit more info about mass spec detection, e.g. about accuracy of determination (ppm or Daltons for MS and MS/MS)

Wish you luck in your professional work. Kind regards

Reviewer 2 Report

The paper addresses a topic worthy if investigation, but it cannot be published in its present form due to some issues connected to data treatment and analysis.

Major issues

l. 91-92: were the microorganisms grown in lab media before experiments? Using frozen microorganisms could significantly affect their resistance to amicrobial compounds

l. 97-98: Which was the amount of DMSO?

l. 105-108: This section is unclear. Please explain counting was done

l. 113-114: Statistic is missing and this is a major drawback

l. 171-172: perform Statistic. I suggest MANOVA (microorganisms and kind of extracts could be used as predictors)

Minor issues:

l. 56/57: I think there is a mistake, because this table is placed on p. 7, l. 173

l. 89: It is table 1

l. 125: It is table 2

l. 171/172: table3 and table 4 

Reviewer 3 Report

Review of the article: “Antimicrobial activity of propolis from the Brazilian stingless bees Melipona quadrifasciata anthidioides and Scaptotrigona depilis (Hymenoptera, Apidae, Meliponini)

Submission ID microorganisms-2050867

Review of the first version is presented below

The manuscript is interesting and generally well prepared. Below I have presented detailed comments and critical remarks. I would be grateful if the authors could answer my questions and take into account my suggestions preparing the final version of the publication.

Detailed comments:

Abstract – some most important results (e.g. MIC/MBC values) should be presented in abstract

Introduction – This part of manuscript quite well presented prepared

Materials and methods

Line 63 – usually 70% ethanol (V/V) is used as a solvent for preparing propolis extract. The authors have not written if they observed any precipitate (insoluble compounds of propolis). From my experience about 30% of the mas of propolis (raw material collected by Apis mellifera) is not soluble in ethanol. Usually the procedure of extract preparing contains the following steps: 1. incubation of the raw material with ethanol (extraction); 2 filtration or centrifugation (separation of the extract and not soluble compounds of propolis); 3. ethanol evaporation; 4. determination of the mass of compounds of propolis that were extracted; 5. solubilisation of these compounds of propolis in the next portion of ethanol to the certain concentration of the propolis.

If some compounds of propolis were not solubilized the authors were not able to say what is the concentration of the propolis in mg/ml.

Line 96 – Mueller Hinton Broth and RPMI media are recommended for determination of MIC values for bacteria and yeasts, respectively.

Line 98 “followed by serial dilution.” – what kind of serial dilution, moreover the investigated  range of concentration of propolis is not given by the authors.

Line 89 - I do not understand why this is Table 4 and table presented on pages 4 and 5 is Table 1?

Results

I am surprized that so many compounds remained unknown (Table 1)

Table 2 – general comment – the observed values of MIC and MBC are very high (e.g. in the case of yeasts they are higher than 10 000 µg/mL). Two more comments – above I have asked about the method of determination of propolis concentration in the extract (what about the precipitate), what kind of serial dilutions were prepared and the investigated range of propolis concentration must be given.

Lines 164-170 – the authors should present the names of microorganisms with italic.

Discussion

The discussion is very short and many important information from literature has been omitted. Some authors, e.g. Gucwa et all., 2018 and Grecka et all., 2019, have found that some fractions of flavonoids are crucial for antimicrobial potential of propolis – what is the authors’ opinion.

In my opinion the discussion is the weakest part of the manuscript and should be importantly improved.

Final decision – major revision.

Round 2

Reviewer 2 Report

Authors addressed all issues

Reviewer 3 Report

The authors importantly improved the quality of the manuscript and answered all my questions and suggestions. In y opinion the current version of the article can be accepted for publication.